

# Effects of the CYP3A inhibitors, voriconazole, itraconazole, and fluconazole on the pharmacokinetics of osimertinib in rats

Yutao Lou[1,2], Feifeng Song[2], Mengting Cheng[2], Ying Hu[2], Yitao Chai[2], Qing Hu[2], Qiyue Wang[2], Hongying Zhou[3], Meihua Bao[4], Jinping Gu[1] and Yiwen Zhang[1,2,5]

[1] College of Pharmacy, Zhejiang University of Technology, Hanghzhou, Zhejiang, China

[2] Clinical Pharmacy Center, Department of Pharmacy, Zhejiang Provincial People's Hospital, Affiliated People's Hospital, Hangzhou Medical College, Hangzhou, Zhejiang, China

[3] Department of Oncology, Zhejiang Provincial People's Hospital, Affiliated People's Hospital, Hangzhou Medical College, Hangzhou, Zhejiang, China

[4] Hunan Key Laboratory of the Research and Development of Novel Pharmaceutical Preparations, Changsha Medical University, Changsha, Hunan, China

[5] Key Laboratory of Endocrine Gland Diseases of Zhejiang Province, Zhejiang Provincial People's Hospital, Affiliated People's Hospital, Hangzhou Medical College, Hangzhou, Zhejiang, China

Corresponding author
Yiwen Zhang,
zhangyiwen@hmc.edu.cn

## ABSTRACT

**Background**. Osimertinib, as third-generation epidermal growth factor receptor tyrosine kinase inhibitor (EGFR-TKI), is the first-line treatment approved to treat advanced T790M mutation-positive tumors. Triazole antifungals are therapeutic drugs for cancer patients to reduce the risk of opportunistic fungal infections. Our objective was to investigate whether three triazole antifungals (voriconazole, itraconazole, and fluconazole) could change the pharmacokinetics of osimertinib in rats.

**Methods**. The adult male Sprague-Dawley rats were randomly divided into four groups ($n = 6$): control (0.3% CMC-Na), and voriconazole (20 mg/kg), itraconazole (20 mg/kg), or fluconazole (20 mg/kg) combined with osimertinib (10 mg/kg) group. Tail vein blood samples were collected into heparin tubes at various time points within 0–48 h after osimertinib administration. Osimrtinib's plasma concentration was detected using HPLC-MS/MS system equipped with a Waters XBridge $C_{18}$ column, with the mobile phase consisting of acetonitrile and 0.2% formic acid water at a flow rate of 0.5 mL/min.

**Results**. Co-administration with voriconazole or fluconazole increased the $C_{max}$ of osimertinib by 58.04% and 53.45%, respectively; the $AUC_{0-t}$ increased by 62.56% and 100.98%, respectively. However, when co-administered with itraconazole, the $C_{max}$ and $AUC_{0-t}$ of osimertinib only increased by 13.91% and 34.80%, respectively.

**Conclusions**. Our results revealed that the pharmacokinetics of osimertinib were significantly changed by voriconazole and fluconazole in rats, whereas it was slightly affected by itraconazole. This work will contribute to a more comprehensive understanding of the pharmacokinetic properties of osimertinib when co-administered with triazole antifungals.

## INTRODUCTION

Drug–drug interactions (DDIs) play a substantial role in the increased risk of adverse drug reactions (ADRs), accounting for approximately 30% of all reported ADRs, which may consequently influence health outcomes (*Bechtold & Clarke, 2021*; *Iyer et al., 2014*). The occurrence of DDIs increases when multiple medications are co-administered. Enzyme inhibition caused by DDIs is of great interest to academic researchers and the pharmaceutical industry. Inhibition or induction of cytochrome P450 (CYP) expression can subsequently lead to DDIs. More than 75% of drug metabolism is based on the CYP superfamily; CYP3A is the most abundant CYP in human physiology and the most critical enzyme in drug metabolism (*Jänne et al., 2015*; *Xiao et al., 2021*).

Patients with cancer are at considerable risk of DDIs. Osimertinib (OSIM) is a third-generation tyrosine kinase inhibitor (TKI) that targets the epidermal growth factor receptor (EGFR) (*Aredo et al., 2021*; *Leonetti et al., 2019*; *Soria et al., 2018*). It is highly selective for patients with advanced non-small cell lung cancer (NSCLC) who have the EGFR T790M positive mutation (*Field et al., 2022*; *Liang, Zhong & He, 2021*). OSIM is predominantly metabolized by CYP3A following oral treatment, which plays a major role in several DDIs (*Ying et al., 2020*). Alterations to the activity of the CYP3A enzyme may cause significant effects on drug exposure; if CYP3A is inhibited and therefore less available, the concentration of other drugs used concomitantly may be increased *in vivo*. Voriconazole (VCZ), itraconazole (ICZ), and fluconazole (FCZ) are common triazole antifungals with moderate to strong CYP3A inhibiton (*Nivoix et al., 2008*).

In clinical practice, as a therapeutic strategy for patients with cancer, OSIM is administered in conjunction with CYP3A inhibitors to decrease the incidence of invasive fungal infections (*Gerber et al., 2020*; *Pagano et al., 2012*; *Vishwanathan et al., 2018*). DDI, however, can be induced, limiting the therapeutic benefits and potentially causing significant adverse effects. Thus, it is necessary to investigate whether CYP3A inhibitors have any effect on the pharmacokinetics of OSIM, due to the possibility of DDI. Therefore, the aim of our study is to evaluate the potential DDIs between OSIM and VCZ, ICZ, and FCZ in rats employing a bioanalytical method that follows the FDA guidance for bioanalytical method validation (*Lou et al., 2022*; *Tang et al., 2020*; *Xu et al., 2019*).

## MATERIALS & METHODS

### Materials

OSIM (≥98.0%) was purchased from the National Institute for Food and Drug Control (Beijing, China). VCZ (≥98.0%), ICZ (≥98.0%), and FCZ (≥98.0%) were provided by Zhejiang Provincial People's Hospital (Hangzhou, China). Nilotinib (≥99.0%), carboxymethyl cellulose sodium, isopropanol, and formic acid of LC-MS grade were purchased from Aladdin Industrial Corporation (Shanghai, China). Acetonitrile and
methanol were purchased from Merck (HPLC grade; Darmstadt, Germany). Ultra-pure water was produced with a Millipore Synergy system (Merck, Darmstadt, Germany). The rat tail vein fixator (Yuyan, China) was provided by the Laboratory Animal Center of Zhejiang Provincial People's Hospital.

## Calibration curve, quality control, and internal standard samples

Two separate stock solutions of OSIM, 0.5 mg/mL, were produced in methanol and stored at −40 °C. The first stock solution was used to prepare working solutions for the calibration curve by serial dilution in methanol at concentrations ranging from 20 ng/mL to 5000 ng/mL. Calibration curve samples consisting of eight non-zero concentrations were obtained by spiking aliquots of 90 µL blank plasma with 10 µL working solutions of calibration solutions (90:10, v/v). Quality control (LLOQ; low, medium, and high QC) samples were prepared in the same manner at concentrations of 20,50, 2000, and 4000 ng/mL. Internal standard (IS) nilotinib was prepared in acetonitrile at a final concentration of 10 ng/mL.

## Chromatography and mass spectrometry

A liquid chromatography with tandem mass spectrometry (HPLC–MS/MS) system coupled to a Jasper™ HPLC and Triple Quad™ 4500MD and equipped with an electrospray ionization (ESI) interface source (AB Sciex, Redwood City, CA, USA) was used for the analysis. The separation was achieved on a Waters XBridge $C_{18}$ C18 column (3.5 µm, 2.1 × 100 mm; Waters Corp., Milford, MA, USA) maintained at 45 °C. Mobile phase A was 0.2% formic acid in water, and mobile phase B was acetonitrile. The gradient elution procedure was optimized as follows: 20% B (0–0.5 min), 20–40% (0.5–1.0 min B), 40–90% B (1.0–2.0 min), 90% B (2.0–3.0 min), and 90–20% B (3.1–3.2 min) at a flow rate of 0.50 mL/min for 4.0 min. Carry-over was minimized with methanol-acetonitrile-isopropanol-water solution (1:1:1:1, v/v/v/v) before and after the injection procedure.

The source parameters were set as follows: ion spray voltage: 5.5 kV, source temperature: 550 °C, GS1: 50psi, GS2: 55 psi, curtain gas: 30 psi, collision gas: 9 psi, entrance potential: 10 V. Multiple reaction monitoring (MRM) was performed in the positive ion mod, and monitoring was carried out for two ion pairs: $m/z$ 500.0 → $m/z$ 72.1 for OSIM (DP = 153 V, CE = 35 V), and $m/z$ 530.0 → $m/z$ 289.1 for IS (DP = 151 V, CE = 41 V). The MS/MS spectra of the ion fragments are illustrated in Fig. 1. The manually tuned MS parameters are provided in Table S1. Data acquisition and quantification were carried out with Analyst™ MD 1.6.3 and MultiQuant™ MD 3.0.2 software (AB Sciex, Redwood City, CA, USA), respectively.

## Sample preparation

The 100 µL of plasma samples (calibration curve, quality control, and rat plasma samples) were precipitated with 100 µL IS (10 ng/mL) and 200 µL acetonitrile. The blank plasma samples were prepared by mixing 100 µL blank plasma and 300 µL acetonitrile. Subsequently, all samples were vortex-mixed for 30 s and were centrifuged at 14,000 × $g$ for 10 min. Finally, each sample was reconstituted with an aliquot of mobile phase A.

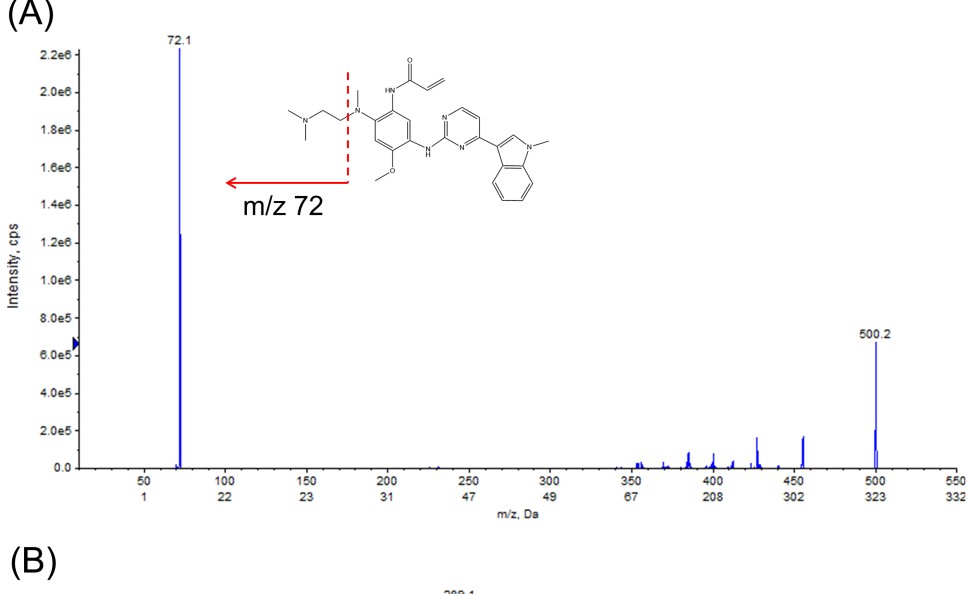

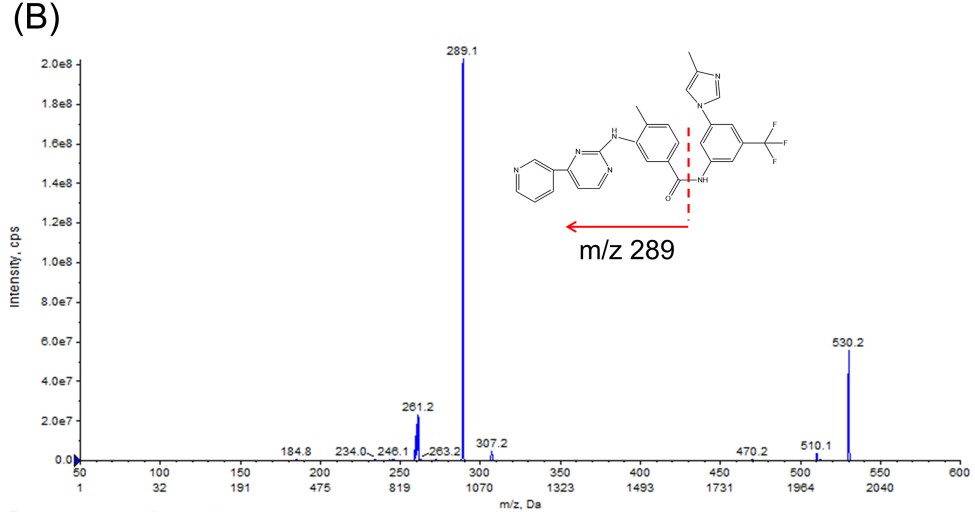

**Figure 1** **Structures and product ion spectra of the protonated molecule [M+H]⁺ ions.** (A) Osimertinib (OSIM) and (B) nilotinib (IS).

The sample was transferred to a 0.8 mL 96-well plate, of which 5 μL was injected into HPLC–MS/MS system for analysis.

## Pharmacokinetic study

Twenty-four adult male Sprague-Dawley rats (220 ± 20 g) with specific pathogen free were supplied by the Laboratory Animal Center of Zhejiang Provincial People's Hospital. All rats were healthy and did not involve genetic modification status, genotype, and any previous procedures. The rats were housed at an animal laboratory (temperature 24 ± 2 °C, relative humidity 50 ± 10%, 12-hour light-dark cycle) for one week as an adaptation period. The study was performed in accordance with the Animal Care and Use Committee of Zhejiang Provincial People's Hospital (number: A20220030).

OSIM, VCZ, ICZ, and FCZ were prepared in 0.3% carboxymethyl cellulose sodium (CMC-Na). After 12 h of fasting and free water consumption, the rats were randomly separated into four groups, and the sample size was based on a similar study using as few rats as possible ($n = 6$): control (equivalent volume of 0.3% CMC-Na), VCZ (20 mg/kg), ICZ (20 mg/kg), and FCZ (20 mg/kg) groups. Only those responsible for treatment management researchers know this group well. Each group was housed in a single cage. The rats in the four groups were orally dosed once daily with 0.3% CMC-Na, VCZ, ICZ, or FCZ, respectively; they reached steady-state exposure after five continuous days of dosing. Thirty minutes after the last administration, each rat received a single oral dose of 10 mg/kg of OSIM. All rats received an oral gavage dose of approximately equivalent volume treatment regimens (one mL). Rats were restrained on the rat fixator on the laboratory bench. Blood samples (300 μL) from the tail veins were collected in heparin tubes at 0.25, 0.5, 1, 2, 3, 4, 5, 6, 8, 12, 24, and 48 h after OSIM administration. Plasma samples were then separated by centrifugation ($3,500 \times g$) for 10 min, and the resultant plasma was frozen at $-30\,°C$ prior to analysis.

### Data analysis

Experimental data were given as mean ± standard deviation (SD). The mean concentration levels of OSIM *versus* time profile were drawn using GraphPad Prism 9.4 (GraphPad Software, Inc., San Diego, CA, USA). Non-compartmental pharmacokinetic parameters of concentration–time data were processed by PKSolver 2.0 (China Pharmaceutical University, Nanjing, China), and the NCA Extravascular module was employed to derive pharmacokinetic parameters for each treatment separately, including $C_{max}$, $AUC_{0-t}$, $AUC_{0-inf}$, $T_{max}$, $t_{1/2}$, $MRT_{0-inf}$, $V_z/F$, and Cl/F. SPSS 26.0 was used for statistical analysis (IBM, Armonk, USA). Significant differences among the groups were identified using one-way ANOVA and Tukey test. $P < 0.05$ was considered significant.

## RESULTS

### Method validation

The chromatographic conditions were carefully optimized to provide the best peak shape and highest response. The retention times were 1.82 and 2.08 min for OSIM and IS, respectively. Representative chromatograms are presented in Fig. 2. The calibration curve was linear from 2 to 500 ng/mL, with correlation coefficients ($r^2$) of over 0.99.

Intra- and inter-day precision and accuracy were measured in sextuplicate at the LLOQ, LQC, MQC, and HQC levels, on three consecutive days. The precision (coefficient of variation, %CV) and accuracy (relative error, %RE) for all QC samples should be within ± 15% (± 20% for LLOQ). For all QC samples, the average %RE ranged from 88.45 to 110.41%, and the intra- and inter-day %CV was less than 7.73% (Table S2).

Extraction recovery and matrix effect were determined using three concentrations of QC samples in five replicates. The peak area ratio of QC obtained in blank plasma pre-extraction was compared to that prepared in samples spiked post-extraction for extraction recovery. The matrix factor was estimated by comparing the peak area ratio of QC generated in samples spiked post-extraction to that of pure solutions at the same

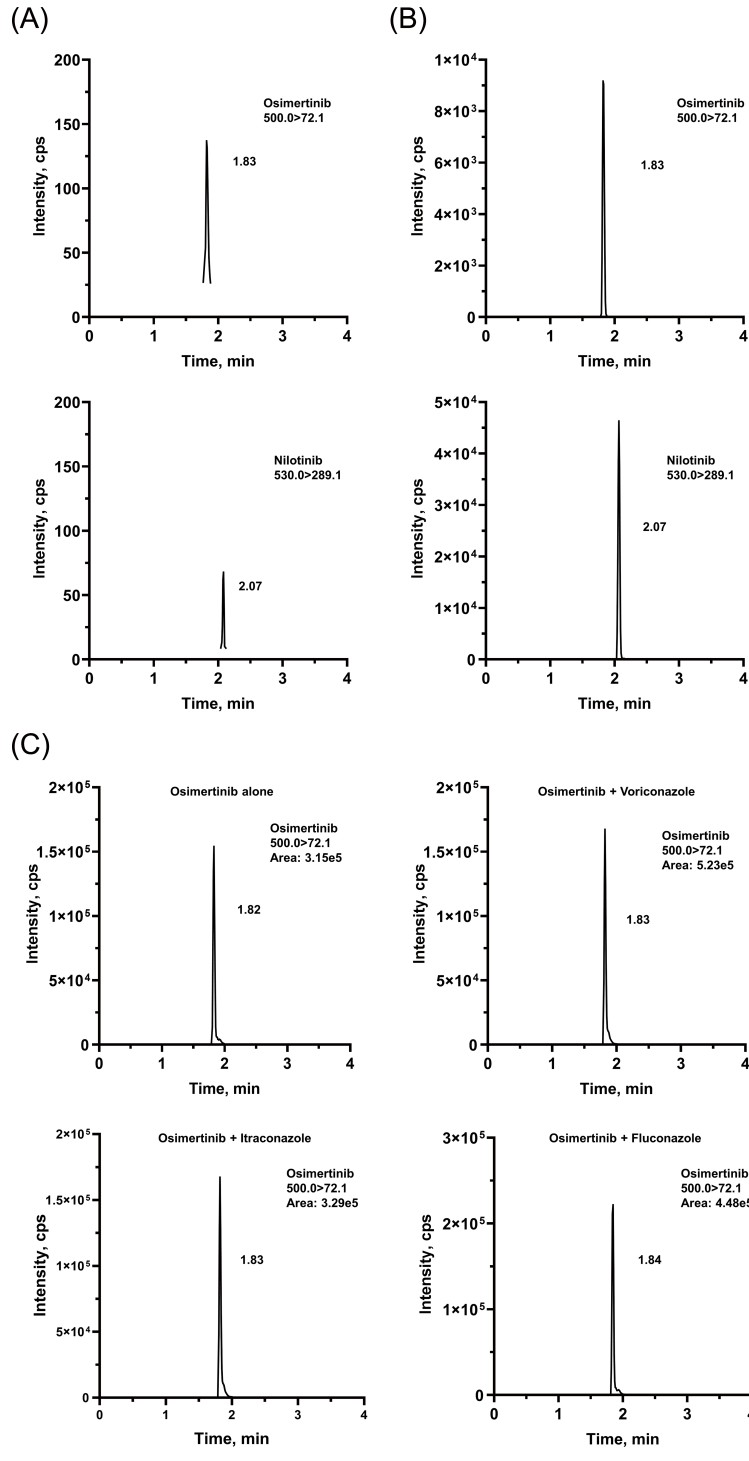

**Figure 2  Representative chromatograms of OSIM and IS.** (A) Blank plasma sample, (B) plasma sample spiked at the concentration of LLOQ for the analyte, and (C) rat plasma sample 4 h after oral administration of OSIM (10 mg/kg).

**Table 1  Pharmacokinetic parameters after oral administration of OSIM (10 mg/kg) (Mean ± SD, $n = 6$).**

| Parameters | Unit | Control | Voriconazole | Itraconazole | Fluconazole |
|---|---|---|---|---|---|
| $t_{1/2}$ | h | $6.65 \pm 0.48$ | $3.82 \pm 0.34$**** | $4.96 \pm 1.10$** | $5.75 \pm 0.25$ |
| $T_{max}$ | h | $2.33 \pm 0.52$ | $2.83 \pm 0.41$ | $3.83 \pm 1.60$ | $3.17 \pm 0.98$ |
| $C_{max}$ | ng/mL | $33.08 \pm 3.66$ | $52.28 \pm 10.47$* | $37.68 \pm 11.53$ | $50.76 \pm 8.41$* |
| $AUC_{0-t}$ | ng/mL*h | $259.45 \pm 33.08$ | $421.77 \pm 58.41$** | $349.73 \pm 96.32$ | $521.44 \pm 92.51$**** |
| $AUC_{0-inf}$ | ng/mL*h | $263.59 \pm 34.44$ | $422.29 \pm 57.90$** | $356.66 \pm 99.68$ | $523.61 \pm 92.44$**** |
| $MRT_{0-inf}$ | h | $7.61 \pm 0.74$ | $6.83 \pm 0.36$ | $7.51 \pm 0.6$ | $8.32 \pm 0.47$ |
| $V_z/F$ | L/Kg | $367.78 \pm 36.26$ | $132.27 \pm 17.94$**** | $227.73 \pm 127.10$** | $162.20 \pm 26.73$*** |
| $Cl/F$ | L/h/Kg | $38.61 \pm 6.21$ | $24.04 \pm 3.12$** | $30.26 \pm 9.89$ | $19.50 \pm 2.71$*** |

Notes.
*$P < 0.05$.
**$P < 0.01$.
***$P < 0.001$.
****$P < 0.0001$.

nominal concentration. The mean extraction recovery of OSIM ranged from 96.25 to 100.67%, with the %CV $\leq$ 2.50% (Table S3). The IS-normalized matrix effect values for OSIM ranged from 97.26 to 105.48% and the %CV was not higher than 4.69% (Table S3).

The stability of the method was evaluated by determining the concentrations of LQC, MQC, and HQC; the evaluation was performed in quadruplicate. Bench-top stability was tested by placing the pretreatment samples at room temperature ($-25$ °C) for 6 h, autosampler stability was assessed by placing the prepared sample in the injector for 24 h, freeze-thaw stability was determined to be adequate over at least three cycles (from $-30$ °C to 25 °C), and long-term stability was assessed to be at least one month before sample preparation (at $-30$ °C for 30 d). For all stability results, the average %RE of OSIM ranged from $-11.04$% to 7.48% across the three QC concentrations, with the %CV less than 6.42% (Table S4).

Carry-over was evaluated by injecting four blank plasma samples over the upper limit of quantitation (ULOQ, 500 ng/mL). The peak area of the blank plasma sample should not exceed 20% of that recorded at LLOQ for OSIM and should be less than 5% of IS. The calculated carry-overs for OSIM and IS were 4.44% and 0.11%, respectively (Table S5).

The above results demonstrated that the assay met the required standards and can be applied to the pharmacokinetic study of OSIM.

### Effect of VCZ, ICZ, and FCZ on the pharmacokinetic profile of OSIM

The mean concentration–time profiles of OSIM alone and OSIM combined with VCZ, ICZ, and FCZ are presented in Fig. 3. The main pharmacokinetic parameters for OSIM are listed in Table 1. The results of the Tukey test are represented in Fig. S1.

These results revealed significant differences in pharmacokinetic parameters between the control and the VCZ or FCZ groups, especially for the values of $C_{max}$, $AUC_{0-t}$, $AUC_{0-inf}$, $V_z/F$, and $Cl/F$ ($P < 0.05$). Concomitant administration of OSIM with VCZ demonstrated an increase in $C_{max}$ and $AUC_{0-t}$ of OSIM by 58.04% and 62.56%, respectively compared with those of administration of OSIM alone. Conversely, the $Cl/F$ of OSIM decreased by 37.74% in VCZ group. The $T_{max}$ value of OSIM in the VCZ group increased by 17.67%,

(A)

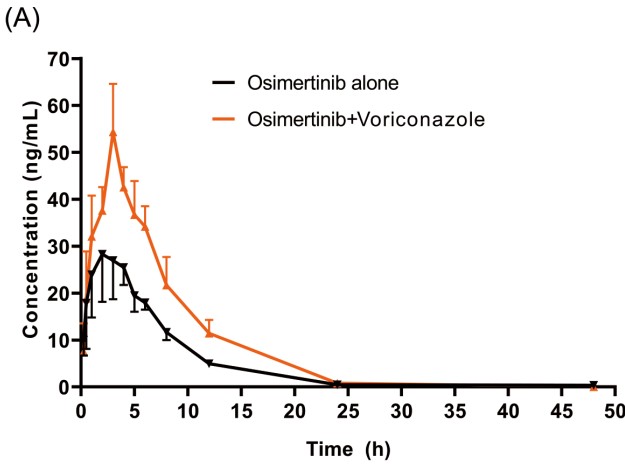

(B)

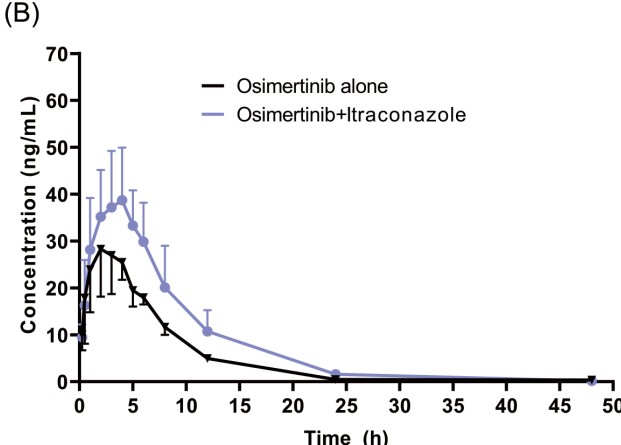

(C)

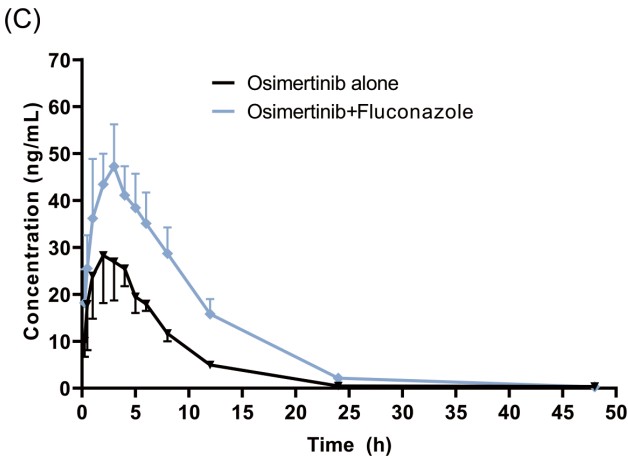

**Figure 3   Mean concentration-time profiles of OSIM.** OSIM combined with (A) VCZ, (B) ICZ, and (C) FCZ (Mean ± SD, $n = 6$).

but the $t_{1/2}$, $MRT_{0-inf}$, and $V_z/F$ values were decreased by 42.56%, 10.25%, and 64.03%, respectively compared with the control. These results implied that VCZ contributed to enhanced systemic exposure and decreased clearance of OSIM. When OSIM was co-administered with FCZ, the Cl/F value of OSIM in the FCZ group decreased by 0.49-fold, but the values of $C_{max}$ and $AUC_{0-t}$ were enhanced by approximately 0.53- and 1.01-fold, respectively. In contrast, co-administration with OSIM and FCZ did not significantly alter the $t_{1/2}$ and $T_{max}$ of orally administered OSIM. Following the administration of OSIM with ICZ, the $t_{1/2}$ and $V_z/F$ of OSIM decreased by 25.41% and 38.08%, respectively ($P < 0.05$), but other parameters ($C_{max}$, $T_{max}$, AUC, MRT, and Cl/F) were not significantly different compared with the control group.

## DISCUSSION

Immunocompromised patients with lung cancer are more vulnerable to invasive fungal infections than healthy individuals (*Gerber et al., 2020*; *Park et al., 2021*). Triazole antifungals, such as CYP3A inhibitors, are frequently used in these patients to prevent fungal infections (*Chen, Li & Chen, 2022*). As a result, they are at a higher risk of experiencing side effects caused by DDIs. However, not all DDIs can be foreseen, and those that can be are not necessarily preventable. Co-administration of OSIM with drugs that are CYP3A substrates may alter the pharmacokinetic profile of OSIM, which could affect its efficacy and/or safety. OSIM is an effective targeted agent for the treatment of the EGFR T790M mutation in patients with NSCLC (*Tang et al., 2019*; *Zhao et al., 2022*). It is primarily metabolized by CYP3A, making it susceptible to metabolic DDIs (*Peters, Zimmermann & Adjei, 2014*). Therefore, we investigated the effect of a strong CYP3A inhibitor, ICZ, and two moderate CYP3A inhibitors, VCZ and FCZ, on the pharmacokinetics of OSIM in rats, because co-administration of CYP3A inhibition may decrease the activity of the enzyme and may increase the blood concentration or decrease the clearance of OSIM.

In this study, the co-administration of OSIM with VCZ, ICZ, and FCZ increased the systemic exposure of OSIM with a 1.14- to 1.58-fold increase in $C_{max}$ and a 1.35- to 2.01-fold increase in $AUC_{0-t}$. VCZ and FCZ have been recommended as CYP3A inhibitors in clinical studies (*Poggesi et al., 2020*). Many studies have shown that CYP3A inhibitors can inhibit the clearance of CYP3A-metabolized drugs. Xun et al. revealed that VCZ inhibited the metabolism of atorvastatin, and had a significant interaction with atorvastatin in patients with fungal infections and dyslipidemia (*Xun et al., 2023*). A recent study demonstrated that VCZ and FCZ increased the systemic exposure of almonertinib by 2.7-fold and 3.4-fold, respectively (*Fu et al., 2023*). Midazolam is regularly utilized as a probe substrate to assess CYP3A activity, and the $AUC_{0-360}$ of midazolam was significantly increased (133%) (*U.S. Food and Drug Administration, 2017*; *European Medicines Agency, 2012*; *Uchida, Tanaka & Namiki, 2014*). Another report indicated that VCZ and FCZ have considerable influence on the formation of 1-OH-midazolam by suppressing CYP3A enzyme activity (*Luo et al., 2019*). Therefore, the increased exposure of OSIM may be due to the inhibitory effect of VCZ and FCZ on CYP3A activity, which greatly affects the elimination of OSIM. Moreover, the Cl/F and the $V_z/F$ were reduced by 37.74% and 64.03%, respectively, in the VCZ group,

and by 49.49% and 55.89%, respectively, in the FCZ group, indicating that the inhibitory effect of VCZ and FCZ on CYP3A reduced the metabolism of OSIM and increased its bioavailability.

Likewise, ICZ has also been recommended as a potent CYP3A inhibitor in clinical DDI investigations (*Liu et al., 2016*; *Prieto Garcia et al., 2018*). Theoretically, the inhibition of CYP3A would increase the plasma exposure of OSIM, especially in combination with a strong inhibitor, such as ICZ. As reported in the literature, ICZ has a significant inhibitory effect on the metabolism of ivosidenib in rats (*Xie et al., 2020*). Another human study showed that co-administration of giretinib with ICZ resulted in a significant inhibition of the clearance giretinib (*James et al., 2020*). Interestingly, ICZ has previously been reported to have a weak effect on the pharmacokinetics of sunitinib and apatinib, both of which are primarily metabolized by CYP3A (*Liu et al., 2018*; *Lou et al., 2019*; *Wang et al., 2021*). In addition, a clinical study observed the AUC and $C_{max}$ of capmatinib alone were similar to those of the combination of capmatinib and ICZ, and that ICZ had no significant effect on the formation of the product CMN288 (*Cui et al., 2023*). Thus, DDI caused by CYP3A-metabolized drugs may be complex and multifactorial, such as experimental design as well as species and ethnic differences. In our study, although co-administration of OSIM and ICZ increased the exposure of OSIM, the main pharmacokinetic parameters, such as $C_{max}$ and $AUC_{0-t}$, were not significantly different between the control and ICZ groups. Furthermore, $T_{max}$ was similar, and the Cl/F was also similar for the two groups, suggesting that the observed increase in exposure was related to an increase in absorption rather than a change in hepatic clearance. Therefore, we concluded that ICZ had no significant effect on the pharmacokinetic profile of OSIM. Our findings were consistent with those of an earlier study assessing the effect of ICZ on the pharmacokinetics of OSIM, in which the $AUC_{0-t}$ was increased by approximately 8% compared with the case wherein OSIM was administered alone (*Brown et al., 2017*).

## CONCLUSIONS

In conclusion, a detailed understanding of the potential for DDIs between OSIM and VCZ, ICZ, and FCZ was gained in the present study. Both VCZ and FCZ had a significant effect on OSIM exposure in rats; however, the inhibition of OSIM metabolism by the strong inhibitor ICZ had no significant effect on plasma exposure. Further human clinical trials of the effects of VCZ, ICZ, and FCZ on OSIM need to be investigated to confirm the significance and accuracy of the interactions.

## ACKNOWLEDGEMENTS

The authors appreciate the contributions of Weijiao Fan for animal experiments.

### Funding

This work was supported by funding from the Zhejiang Provincial Natural Science Foundation of China (Grant No.: LYQ20H310001, LYY21H310011), the Chinese Medicine Research Program of Zhejiang Province (No.: 2021ZZ001), and the Medical and Health Research Program of Zhejiang Province (Grant Nos.: 2021KY040, 2022KY069). The funders had no role in study design, data collection and analysis, decision to publish, or preparation of the manuscript.

### Grant Disclosures

The following grant information was disclosed by the authors:
Zhejiang Provincial Natural Science Foundation of China: LYQ20H310001, LYY21H310011.
Chinese Medicine Research Program of Zhejiang Province: 2021ZZ001.
Medical and Health Research Program of Zhejiang Province: 2021KY040, 2022KY069.

### Competing Interests

The authors declare there are no competing interests.

### Author Contributions

- Yutao Lou conceived and designed the experiments, performed the experiments, analyzed the data, authored or reviewed drafts of the article, and approved the final draft.
- Feifeng Song performed the experiments, analyzed the data, authored or reviewed drafts of the article, and approved the final draft.
- Mengting Cheng conceived and designed the experiments, performed the experiments, authored or reviewed drafts of the article, and approved the final draft.
- Ying Hu analyzed the data, prepared figures and/or tables, and approved the final draft.
- Yitao Chai performed the experiments, analyzed the data, authored or reviewed drafts of the article, and approved the final draft.
- Qing Hu conceived and designed the experiments, authored or reviewed drafts of the article, and approved the final draft.
- Qiyue Wang analyzed the data, prepared figures and/or tables, and approved the final draft.
- Hongying Zhou performed the experiments, prepared figures and/or tables, and approved the final draft.
- Meihua Bao analyzed the data, prepared figures and/or tables, and approved the final draft.
- Jinping Gu analyzed the data, authored or reviewed drafts of the article, and approved the final draft.
- Yiwen Zhang conceived and designed the experiments, analyzed the data, authored or reviewed drafts of the article, and approved the final draft.

## Animal Ethics

The following information was supplied relating to ethical approvals (*i.e.*, approving body and any reference numbers):

The Animal Care and Use Committee of Zhejiang Provincial People's Hospital approved the study (Number: A20220030).

## Data Availability

Raw data are available in the Supplemental Files.

## Supplemental Information

Supplemental information for this article can be found online at http://dx.doi.org/10.7717/peerj.15844#supplemental-information.

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
