# Peer review of "Effects of the CYP3A inhibitors, voriconazole, itraconazole, and fluconazole on the pharmacokinetics of osimertinib in rats"

_PeerJ, doi:10.7717/peerj.15844_

## Round 0.1 · original submission · Minor Revisions

Please address concerns of both reviewers. Please note that you are not obligated to cite papers recommended by the reviewer. You can do so, if those papers are related to your study. Please also note that any suggestion that the findings from the rodent study are in any way applicable to humans should be deleted from the manuscript. Drug interaction studies done in rodents are in no way predictive of what will happen in humans. It is already well-established in numerous human studies that itraconazole, fluconazole, and voriconazole will inhibit clearance of drugs metabolized by CYP3A. That would need to be fully acknowledged and discussed in the revised manuscript, with citations to pertinent original research appearing in published and indexed biomedical literature

Reviewer 1 ·

Basic reporting

This manuscript was well written to investigate whether three triazole antifungals (voriconazole, itraconazole, and ûuconazole) could change the pharmacokinetics of osimertinib in rats. The title and abstract clearly mention the study goal and are found appropriate.

Experimental design

Although this work is interesting and relevant for the preclinical study, some issues need to be resolved:
1. Why use Nilotinib as IS in this study?
2. The FDA guideline on method validation should be detailed, considering cite the following references.
(1) Xu RA; Lin Q; Qiu X; Chen J; Shao Y; Hu G; Lin G; UPLC-MS/MS method for the simultaneous determination of imatinib, voriconazole and their metabolites concentrations in rat plasma, J Pharm Biomed Anal, 2019, 166: 6-12.
(2) Tang C; Niu X; Shi L; Zhu H; Lin G; Xu RA; In vivo Pharmacokinetic Drug-Drug Interaction Studies Between Fedratinib and Antifungal Agents Based on a Newly Developed and Validated UPLC/MS-MS Method, Front Pharmacol, 2020, 11: 626897.
3. Pharmacokinetic parameters in Table 1 should be detailed as label.
4. The English should be improved.

Validity of the findings

This manuscript was well written to investigate whether three triazole antifungals (voriconazole, itraconazole, and ûuconazole) could change the pharmacokinetics of osimertinib in rats.

Reviewer 2 ·

Basic reporting

This study developed a sensitive and selective LC-MS/MS method for determining osimertinib in rat plasma. The manuscript exposed has scientific and clinical importance for researching effects of CYP3A inhibitors voriconazole, itraconazole and fluconazole on the pharmacokinetics of osimertinib in rats, but there are some problems with this manuscript shown below:
1. Reasonably clear written manuscript in need of a more thorough review from a fluent English speaker. For examples, ‘One hundred microliters’ should be written as 100 μL.
2. What was the solvent to plasma ratio for preparation of calibrators?
3. How were the animals constrained for blood sampling from the tail vein?
4. How was the dosage of the drug administered to rats determined?
5. Please describe if PK parameters are normally distributed. Why was one-way ANOVA used to assess differences between groups?

Experimental design

see above

Validity of the findings

see above

---

## Round 0.2 · accepted · Accept

All concerns of both reviewers are addressed and the manuscript is revised accordingly.